# Use-Specific Considerations for Optimising Data Quality Trade-Offs in Citizen Science: Recommendations from a Targeted Literature Review to Improve the Usability and Utility for the Calibration and Validation of Remotely Sensed Products

**Alice Schacher [1], Erin Roger [1],\*, Kristen J. Williams [1], Matthew P. Stenson [1], Ben Sparrow [2] and Justine Lacey [1]**

[1] Commonwealth Scientific & Industrial Research Organisation (CSIRO), Canberra, ACT 2601, Australia
[2] The Terrestrial Ecosystem Research Network, Adelaide, SA 5005, Australia
\* Correspondence: erin.roger@csiro.au

**Abstract:** The growth of citizen science presents a valuable potential source of calibration and validation data for environmental remote sensing at greater spatial and temporal scales, and with greater cost efficiency than is achievable by professional in situ reference-data collection alone. However, the frequent mismatch between in situ data-quality requirements for remote-sensing-product development and current data quality assurance in citizen science presents a significant challenge if widespread use of these complementary data sources is to be achieved. To evaluate the scope of this challenge, we conducted a targeted literature review into the nature of data-quality issues faced by citizen-science projects for routine incorporation into terrestrial environmental-monitoring systems. From the literature, we identify the challenges and trade-offs to inform best-practice implementation of data quality assurance in citizen-science projects. To assist practitioners in implementing our findings, we grouped these themes by stage of citizen-science project: (1) program planning and design; (2) participant engagement; (3) data collection; and (4) data processing. As a final step, we used our findings as the basis to formulate guiding questions that can be used to inform decision making when choosing optimal data-quality-improvement and assurance strategies for use of citizen science in remote-sensing calibration and/or validation. Our aim is to enhance future development of citizen-science projects for use with remote sensing in environmental monitoring.

**Keywords:** citizen science; remote sensing; data quality; monitoring; calibration; validation

## 1. Introduction

Citizen science is increasingly being applied to scientific methodology, and involves the often-voluntary contributions of generally non-expert 'citizens' towards various processes, including data collection, analysis and hypothesis generation [1]. Citizen science can be broadly defined as public participation in scientific research and knowledge production [2]. Citizen-science projects are already delivering benefits to both science and society in a wide variety of ways [3,4]. The development of citizen-science projects focused on environmental monitoring has already seen citizen scientists collecting a diverse array of data on ecosystems at unprecedented spatial and temporal scales [4,5]. As such, the scale and cost efficiency of environmental data collection by citizen science, offers a promising potential source of complementary data to the large-scale information products generated by remote-sensing technologies. However, the use of citizen-science data with environmental monitoring still presents several challenges, including data interoperability, data quality, participant engagement and issues of privacy [6–9].

A challenge for remote sensing is securing adequate and reliable in situ observation data with which to calibrate and validate remote-sensing products [10,11]. Remote-sensing

activities typically use ground-based data in one of two ways: calibration and validation. Calibration refers to data used to help convert raw remote-sensing data to usable measures of reflectance, whereas validation refers to the use of data to determine the accuracy/precision of products resulting from analysis. These activities give rise to the question of whether citizen science can play a role in generating field data (plot or point-based) from vegetation and ecosystem conditions that can be used to calibrate and/or validate remotely sensed data products [12]. Remotely sensed data provides a very broad spatial and temporal coverage, but the resolution and ability to extract useful indicators such as habitat, vegetation condition and biodiversity, requires ongoing access to data for image training, calibration and validation [13]. Funding limitations can result in professionals needing to compromise on re-visit frequency [14]. As a consequence, this can result in long return-to-site visit times, leading to the need to account for these survey gaps within modelling approaches in order to be able to quantify rapid change and inform management decisions [15].

We aim to explore these challenges and subsequently identify the critical considerations that would enable citizen-science data to be used to perform a fit-for-purpose calibration/validation exercise of remotely sensed products to assist with monitoring ecosystem conditions in terrestrial environments. Here, we synthesise the challenges and recommendations that have been reported in the literature into overarching themes. We then use these themes to generate a series of guiding questions to inform both optimisation and future development of citizen-science projects for remote sensing. By doing so, we show how to optimise data quality trade-offs in citizen science to improve its usability and utility for calibration. Our goal is to identify commonalities among these challenges and recommendations such that they may be applied by practitioners to help meet the requirements of remote sensing and inform future citizen-science program design.

### 1.1. Current State of Citizen-Science Use in Remote Sensing

The research interest in combining citizen science and earth observation has grown rapidly, with a marked increase year on year from 2005 to 2016 in publications mentioning 'citizen science' and 'earth observation' in Scopus [6]. The capacity of citizen science to provide validation and reference data at broader spatial and temporal scales than traditional field surveys, positions it well to complement future calibration and validation efforts [6]. Not only is the potential accessible extent of on-ground data collection expanded, it can be more cost effective than traditional methods alone [6]. However, the information collected by citizen scientists should be seen as complementary to professionals, as the two data sources cannot easily be compared, due to the often extensive training required to match professional levels. For example, a major challenge raised regarding the suitability of citizen science is the lack of standardisation and heterogeneity of participation, which can hinder data quality and/or the quantification of error in citizen-science data [6–8]. Furthermore, the nature of participation of citizen scientists in projects is typically non-uniform, and participants may be involved in any stage, such as data collection, interpretation and authorship [16].

In the context of earth observation, there are two primary means whereby citizen science is already contributing to the calibration and validation of remote-sensing products [17]. First, through the direct validation of land-cover imagery via crowdsourcing classification and verification of images. For example, the Geo-Wiki project [18] used local knowledge from a global network of volunteers to evaluate the accuracy of satellite-derived land-cover maps, and the VIEW-IT project [19] used a team of students and expert moderators to collect reference samples for producing and verifying the accuracy of satellite-derived land-use/land-cover maps. Secondly, citizen scientists may be recruited to source data in situ to provide either supplementary datasets for improving the calibration of remote-sensing products, or ground-based observations that may be used to ground truth remote-sensing products [12].

*1.2. Suitability of Citizen Science for Remote-Sensing Calibration/Validation*

There is diversity in the potential role and design of citizen-science programs to assist remote sensing [20]. Citizen scientists could be engaged to assist in the calibration and validation of remote-sensing products at a number of stages in the data-processing chain, namely: validation and description of remote-sensing data, collection of observations, description, and correlation of data with remote-sensing data [6]. The stage at which citizens are engaged and the nature of data capture by citizen scientists underpin type-specific limitations, research challenges, best practice, and the novel benefits offered by remotely sensed environmental monitoring. Chandler et al. [4] highlight that highly correlated remote-sensing products require specific considerations for optimising data-quality trade-offs in citizen science, and are limited to secondary productivity, nutrient retention, ecosystem extent/fragmentation, and ecosystem functional-type composition. Other variables such as species distribution, population abundance and habitat structure, (while not directly translatable to the validation of remotely sensed products) can be used as supplementary data [4]. However, it is also important to note that the requirements for calibration and validation can be substantially different. For example, citizen-science data might meet the requirements of calibration data, but be unsuitable for validation.

*1.3. Data Quality in Citizen Science: Challenges and Trade-Offs*

While citizen science is well positioned to provide reference and validation data in relatively large quantities and at large spatiotemporal scales, these dimensions reflect only one aspect of data quality [21]. Data quality requires a trade-off between relevant attributes such as data accuracy, data completeness and data timeliness [21,22]. While the capabilities of citizen scientists can be harnessed to generate large volumes of data, much of the reluctance associated with the use of citizen-science data surrounds uncertainty about its accuracy [9,23,24]. For example, while the task of describing remote-sensing data by citizen scientists is possible, accuracy would be difficult to quantify, due to its interpretive nature. This is highly relevant to the prospective applications of citizen-science data in validation practices, wherein standards applied by professional land-survey protocols require sampling-practice consistency to provide ground-truthing data [25]. Therefore, if data is to be usable it must first be of trustable and usable quality, and supported by methods developed to verify observer reliability [26].

## 2. Methods

To identify the opportunities and challenges of using citizen-science data for calibrating and validating data for remote sensing of the environment, we conducted a targeted literature review, as per Ceccaroni et al. [27], into the data-quality issues faced by citizen science to better understand how to increase uptake in earth observation [28]. We did this using combinations of key search terms and Boolean operators. The following databases were included in our search: Scopus and Web of Science. The sources comprise peer reviewed literature from different disciplines (e.g., environmental sciences and remote sensing). We employed both, to obtain more reliable and cross-checked data. The initial search terms were 'citizen science' and 'data quality' and 'remote sensing' and were restricted to between January 2011 and December 2022, using the 'published from' field. This search yielded 61 results for 'citizen science' and 'data quality' and 136 results for 'remote sensing'. We then screened the papers for duplicates within each database, and removed them. We also excluded papers if their subject material was non-generalisable (not easily synthesised) or focused on non-terrestrial environmental data collection such as water and air quality [29]. Additional search terms using combinations of 'crowd sourcing', 'data accuracy', 'remote sensing', 'ecology', 'vegetation', 'data validation' 'volunteered geographic information (VGI)' and 'environment' were subsequently used, based on key words taken from articles from the initial search, and with the recognition that some of these terms are used interchangeably in the literature. A 'snowballing' method from cited literature was also used to identify additional papers. It is important to note that our aim

was not to conduct a systematic literature review, as in, e.g., Vasiliades, Hadjichambis, Paraskeva-Hadjichambi, Adamou and Georgiou [28], but rather a targeted review, so as to thematically present an overview of the current state of data-quality issues and identify appropriate considerations for data-quality improvement and assurance for citizen-science data use in remote sensing in terrestrial environments.

To present our findings, we first grouped results by the relevant stage of citizen-science project, e.g., planning and design through to data processing and evaluation. Stages used in this review were adopted from those presented in the literature [2,6]. This decision was made to assist practitioners in more easily implementing our findings. Subcategories within each stage of citizen-science projects were found based, on our approach of listing key generalisable themes using a semi-structured checklist including a matrix of data-quality requirements. All relevant information from the literature was identified, grouped and named [2,6]. A summary of our findings from the literature, highlighting tips for programmatic enhancements and considerations is presented in tables below each section.

Our last step was to present key guiding questions associated with each stage of a citizen-science project as well as the relevant subcategories. Questions were formulated from the literature and were decided on based on discussion and agreement among the authors (two of whom have extensive experience with the design and implementation of citizen-science projects and four of whom are scientific experts in environmental monitoring and remote sensing). Our basis for this discussion was a list of generalisable themes arising from the section above, again generated using a semi-structured checklist to guide commonalities.

## 3. Results

There was consensus across the literature that citizen-science programs aimed at producing usable data for remote-sensing purposes need to prioritise the usability of data quality in a project and in an application-specific manner. Trade-offs between dimensions of data quality should prompt resource investment into the elements of program design that will be of most value to the intended end use. The role of technology is a key enabler in improving data quality through data standardisation, such as supporting geotagged images and the use and support of artificial intelligence. Below, we have detailed the data-quality-improvement actions that may take place across multiple stages of citizen-science projects. The summary is based on our review of the literature, noting that we did not run any analyses to test the assertions from the literature presented below. Rather, the findings synthesise and present the current state of the debate in this field across the life cycle of a citizen-science project.

### 3.1. Planning and Design

From the outset, projects must be developed with clear research and engagement goals to inform program design [30]. For use in remote-sensing calibration and/or validation, it is important that programs or post hoc analysis techniques are developed with predetermined levels of acceptable error (i.e., fit for purpose). Depending on the application of citizen-science data, and the type of remotely sensed data, the goals of the use of data may vary considerably. Applicable data types for remote-sensing calibration/validation can be broadly classified as the direct validation/visual interpretation of remote-sensing images, the in situ collection of reference data sets, and the validation/description of in situ data sets [31]. Projects may facilitate remote-sensing calibration and/or validation by applying one or more of these approaches. Credible use of citizen-science data for these purposes will rely not only on meeting minimum accuracy and spatial and/or temporal coverage requirements, but also on effective declaration of error levels and sources [31]. Key considerations affecting data quality in the program planning-and-design stage are outlined below.

(i)    End use of data

Identifying how citizen-science data will contribute to remote-sensing calibration and validation is critical to determining the relative value of prioritising data quality across the program [32,33]. For example, investment might be placed in data-quality refinement or characterisation based on the cost of error in the citizen-science data for its intended use. These uses might be contributing to the direct validation of remotely sensed images or to the direct in situ calibration/validation of data, or be indirectly associated with in situ data for the improvement of remote-sensing modelling. Where citizen-science data is integrated with other sources of calibration/validation data, the effect of error in citizen-science data may be moderated through the clear identification and characterisation of error sources and covariates [34,35]. Similarly, the type of data to be collected will influence error rates and suitability towards different applications in remote sensing.

(ii)   Trust, transparency and privacy

The conflict between privacy, transparency and trust is often amplified in citizen science [36]. Clear planning and privacy/transparency standards should be set prior to program implementation and be clearly communicated to participants (for a review of human ethical considerations in the context of citizen science, see Rasmussen and Cooper [37]). Furthermore, the factors influencing participant and user trust should be evaluated in a project-specific and context-specific manner, prior to the development of privacy and data-sharing policies. Iterative evaluations and remediations should be made throughout the program implementation [36]. Key program-enhancement considerations are summarised in Table 1.

**Table 1.** Program planning and design: Programmatic enhancement considerations.

| | |
|---|---|
| End use of data [4,33,34] | • The degree of reliance on the citizen-science data will increase the cost of error in the data, increasing the importance of data quality assurance and accuracy. <br> • Where citizen science is integrated with other sources of data, the effect of error in citizen-science data may be moderated through clear identification and characterisation of error. <br> • Data types to be collected by citizen science are limited by the overlap between data sources translatable between both citizen science and remote sensing. <br> • Research goals, end use and data-quality requirements should be clearly defined a priori, through collaborative engagement of stakeholders from relevant domains. <br> • Program design, protocols and valuation of respective elements of data quality should reflect these project-specific end-use goals. |
| Transparency, trust, and privacy [36] | • The conflict between privacy protection, transparency and trust is often amplified in citizen science, and the importance of data quality and transparency is heightened for use with remote sensing. <br> • Clear planning and privacy/transparency standards should be set prior to program implementation, to be clearly communicated to both participants and stakeholders. <br> • The factors influencing participant and user trust should be evaluated in a project-specific and context-specific manner prior to the development of privacy and data-sharing policies, and iterative evaluations and remediations should be made throughout the program implementation. |

### 3.2. Participant Engagement

The engagement of participants is crucial to the success of programs, and underpins the utility of citizen science for remote-sensing calibration and/or validation applications

through the quantity and diversity of participants. The spatial and temporal extent of engagement is of particular importance to the value offered by citizen science over other means of data collection [38]. The diversity of mass participation and inclusionary practice in citizen science, however, can also amplify uncertainty in the accuracy and sources of bias/error within data. Programs should employ strategies to motivate a diverse range of participants, while collecting participant metadata regarding the diversity of participants and factors of bias, to facilitate program refinement [39]. The optimal number of contributors and the duration of participation may differ between programs, depending on the costs of participant training, the ideal quantity of contributions, the sources of bias and the computational efficiency of the data analysis [40,41]. Key considerations affecting data quality in the participant-engagement stage are outlined below.

(i)     Participant background

It is essential to consider whether participant background will affect data quality [42]. The value of participant background diversity or specificity will be project specific, and local knowledge, for example, may be a valuable source of unique insights. Prior knowledge and skills will also be likely to influence the efficiency of participants in relation to their effort [39]. Training may be one way of mitigating the effects of variability in participant background [38]. Furthermore, having economic, health and conservation stakes in the outcomes of a project can increase participant engagement [43], but such stakes may also bias observations. Researchers should consider evaluating potential conflicts of interest among participants, while considering that these may also offer useful information about the diversity of perspectives and access to private or commercial land (e.g., there may be benefits to engaging landowners or those employed in commercial agriculture or forestry) [44].

(ii)    Training

Participant training presents a useful tool in balancing the effects of participant background diversity. Ongoing evaluation of training effects through feedback and testing is largely recommended to improve training effectiveness, direct training resources and improve understanding of data-quality covariates [20]. The effectiveness of training should be considered not only in relation to data quality, but also in its effect on perceived confidence and engagement of participants [45]. While initial costs may be incurred to produce training data sets, ongoing training data may be provided by high quality contributions. Selective attention may improve efficiency for direct remote-sensing applications (e.g., land-cover type); however, it may reduce the detection of additional data of indirect value to calibration/validation applications (e.g., factors influencing environmental change) or create an effect of learned inattention [46].

For researchers, participant training is seen as a data-quality-improvement strategy and, for participants, training represents a basis for inclusion in the scientific process and in learning [47]. Training is a tool for improving participant engagement if training protocols address these motivational factors. Conversely, arduous or extensive training protocols may risk reducing participant engagement [42,46,48]. Participant training can improve the accuracy of sampling design, namely in structured sampling protocols [39], in standardising methods, increasing participant knowledge and skill, and increasing participant confidence, to increase consistency [20,22,43,45]. However, it has also been noted that overtraining, and high task complexity can reduce the quantity and completeness of contributions, making the best course of action often project dependent.

(iii)   Incentivisation

Directing participant effort towards points of heightened interest to researchers (e.g., landcover-type boundaries), or to dispersed data points for greater coverage, may be beneficial. Gamification or credit-based approaches may present useful tools for efficiently directing participant effort. Participant effort in validating remote-sensing images may be realised through tasking systems that facilitate consensus classification; that is, increasing

the number of observations for difficult or valuable data points. While reward systems, credit-based systems and gamification may strongly incentivise continued participation, gamification could also lead to participants gaming the system, and may also risk disincentivising some participants with altruistic motivations [42].

(iv)   Feedback

Feedback and communication between participants can increase the quantity and completeness of observations through benefiting participant engagement, learning and confidence [6,36,42,45]. However, increased communication among participants may also increase risk of bias, 'group-think', hypothesis guessing and treatment spillover [42]. Embedded feedback systems within task design are recommended for improving participant-contribution completeness and accuracy [45,49]. Some programs may wish to incorporate performance evaluation and/or reputation metrics (i.e., participant-expertise ratings, where applicable), for participants.

(v)   Participant motivation and retention

To optimise the recruitment and retention of participants, programs should consider how motivation can be evaluated [6,39,42,50]. For example, programs might usefully be tailored to the motivations of participants from the outset and throughout the program implementation [33,47]. There may also be a training effect incurred by retention of participants [33,39] and these effects have been observed to increase the competency of contributors. However, the effects may be variable, and participation may not predict contribution quality [51]. Where socialisation among participants occurs, retention of participants can create experienced contributors capable of providing useful feedback and assistance to novice participants. However, long-term participation can cause a participant 'staleness' effect, resulting in poorer-quality contributions regarding completeness and quantity [49]. Key program-enhancement considerations are summarised in Table 2.

**Table 2.** Participant engagement: Programmatic enhancement considerations.

| | |
|---|---|
| Participant background [29,38,42–44] | • The value of participant background diversity or specificity will be project specific; in the context of remote sensing, local knowledge for example, may provide unique insight. <br> • Prior knowledge and skills will likely influence the efficiency of participants in relation to effort. However, further research and project-specific investigations would be required to account for the extent of this effect. <br> • Training may mitigate the effects of variability in participant background <br> • While having economic, health and conservation stakes in the outcomes of a project can increase participant engagement, such stakes may also bias observations. Researchers should consider evaluating conflicts of interest among participants, while considering that these may also offer useful information such as in the diversity of perspective and access to private or commercial land. |
| Training [20,42,45,47] | • Ongoing evaluation of training effects through feedback and testing is largely recommended to improve training effectiveness, direct training resources and improve understanding of data-quality covariates. <br> • The effectiveness of training should be considered not only in relation to data quality but in its effect on perceived confidence and the engagement of participants, through periodic evaluation. <br> • While initial costs may be incurred to produce training data sets, ongoing training data may be provided by high-quality contributions. |

**Table 2.** *Cont.*

| | |
|---|---|
| | • While for researchers, participant training is seen as a data-quality-improvement strategy, for participants, training represents a basis for inclusion in the scientific process and an opportunity for learning.<br>• Arduous or extensive training protocols may risk reducing participant engagement.<br>• Training complexity, like task complexity, should be relatively simple if high participation numbers are required.<br>• The effectiveness of increased training effort may be increased where observer-bias effects are a primary challenge; this may include training participants on how to select suitable sampling sites for collection of in situ data, or training in areas where the cost of misidentification or false-absence data will greatly reduce data usability. |
| Incentivisation [20,47] | • Reward systems, point-based systems and gamification can strongly incentivise continued participation and possibly provide tooling for directing participant effort; however, gamification may risk disincentivising some more valuable participants with altruistic motivations.<br>• Gamification and point-based approaches, in addition to training on sampling-location selection, may be useful tools to efficiently direct participant effort. |
| Feedback [6,39,45,49] | • Contributor confidence and data accuracy is aided by providing practice data-collection opportunities with integrated feedback systems.<br>• Embedded feedback systems within task design are recommended, to improve participant-contribution completeness and accuracy |
| Participant motivation and retention [6,39,42,49,50] | • To optimise the recruitment and retention of participants and maximise their engagement, programs should consider how participant motivation can be directly evaluated.<br>• Participant motivation affects the completeness and quantity of observations.<br>• Lengthy participation may cause a participant 'staleness' effect, resulting in poorer-quality contributions in terms of completeness and quantity. |

### 3.3. Data-Collection Protocols

Program development must consider design choices with respect to associated trade-offs. These trade-offs include deciding on the level of data-collection-protocol standardisation, task difficulty, metadata collection, data-capture requirements and sampling repetition [52]. The literature suggests that subjectively estimated measures are not suited to valid interpretation of data. Such data types include visual-cover estimates that are highly translatable to remote-sensing land-cover and ecosystem functional-type data. While visual-cover estimates are often highly variable among non-experts, variability in these subjective interpretations is also common among professionals [51,53]. It may be useful to generate subjectively interpreted cover estimates for calibration/validation data which may benefit from the aggregation of multiple responses using geo-located photographic data. Geo-located photographic data (due to the resolution) can usefully provide land-cover information (e.g., the Degree Confluence Project, [54]), species presence (e.g., iNaturalist, [44]), and azimuth images at reference points (e.g., GLOBE observer app, [17]), but the sampling strategies should be directed to facilitate the scalability of this information. Key considerations affecting data quality in data collection are detailed below.

(i)     Sampling scale and density

For larger datasets, there is a trade-off between the computational efficiency of citizen-science data analysis and the possible scale of inference [55]. However, it may be possible

to mitigate biases through program design. Spatial and temporal biases are likely to follow common patterns of human movement and population density, and even more so in unstructured citizen-science design, which can greatly influence the variability and density of data [44,55,56]. Patterns of spatial bias increase with variables such as mass participation, opportunistic sampling and untrained-amateur reliance [57]. Where mass participation is sought, or ephemeral participation is likely, spatial-sampling biases can be mitigated in program design and/or integrated with professional data [57].

(ii)    Structure and standardisation of sampling protocols

Where data completeness relies on accurate absence information, or estimation of abundance, structured-sampling protocols may be necessary [51]. However, increasing the structure of sampling may incur costs associated with training and loss of participation, while facilitating simpler data analysis. Unstructured sampling is best suited where presence-only information is used, such as distribution estimation. This is because unstructured sampling will be prone to much lower signal-to-noise ratios. The integration of tools to capture confidence and effort information are of increased importance for programs producing usable, unstructured citizen-science data [38]. The feasibility of reliably standardising data-collection protocols will be reduced with increasing task complexity, and/or the incorporation of subjective estimation [39,51]. Some recommendations that overemphasise reducing functional differences between volunteers and professionals may detract from the inherent quality offered by the unique value proposition of citizen science [48]. The use of mobile applications to standardise data input and improve field data has been shown to reduce curation costs [58].

(iii)   Metadata capture and technology

Efforts to collect metadata such as time of observation and perceived difficulty, provide useful sources of covariate information when estimating certainty in data and potential sources of error [38,59–61]. Collection of participant confidence level can also help calibrate citizen-science data and be used to direct training effort or evaluate the effectiveness of training [51,59,62]. The use of inbuilt sensors in smartphones is growing in popularity. However, the accuracy and capabilities of these devices is variable, and may influence the accuracy of data [42]. These sensors, along with controlled-data-input requirements can reduce the syntactic-error rate in data entry [45]. Issues include how technological requirements may reflect technological availability within the intended pool of participants, and whether these requirements are likely to influence the diversity of participants. However, the use of progressive web applications is helping to resolve many of the issues associated with device compatibility, thus making the issue of technological availability less of a barrier. Key program-enhancement considerations are summarised in Table 3.

**Table 3.** Data collection: Programmatic enhancement considerations.

| | |
|---|---|
| Sampling scale and density [39,40,44,55,57,60] | • For larger datasets, there is a trade-off between the computational efficiency of data analysis and the possible scale of inference.<br>• Spatial and temporal biases are likely to follow common patterns of human movement and population density in citizen science, and even more so in unstructured citizen-science design, which can greatly influence the variability and density of data.<br>• Where mass participation is sought, or ephemeral participation is likely, spatial-sampling biases can be mitigated in program design through the collection of secondary data pertaining to contributor trust and biases to facilitate modelling, and through simplification of sampling protocols in both opportunistic and structured schemes. |

**Table 3.** *Cont.*

| | |
|---|---|
| | • Optimal sampling size and density should be determined in early stages of development, as there may be diminishing returns on data-analysis effort and reductions in signal-to-noise ratio, with larger sample sizes. <br> • Post hoc data-analysis strategies may also be used to correct for some degree of spatial bias, but this will only account for variation in sampling density—if spatial coverage of data is insufficient for remote-sensing purposes, program managers may consider incentivisation of data-poor locations. |
| Structure and standardisation of sampling protocols [38,51] | • Where data completeness relies on accurate absence information, or estimations of abundance, structured-sampling protocols may be necessary. Increasing the structure of sampling may incur costs associated with training and loss of participation while facilitating simpler data analysis. <br> • Unstructured sampling is best suited where presence-only information is used, such as distribution estimation—unstructured sampling will be prone to much lower signal-to-noise ratios and require greater investment in effective data-analysis strategies.The integration of tools to capture confidence and effort information are of increased importance to programs producing usable, unstructured citizen-science data. <br> • For remote sensing, simpler task design and increased sampling standardisation is likely to increase data accuracy and validity for end use when collecting in situ calibration and/or validation data. |
| Metadata capture and technology [38,59–61] | • Effort metrics including time and perceived difficulty provide useful sources of covariate information when estimating certainty in data and potential sources of error. |

### 3.4. Data Processing

Several strategies employed in data collection and program design are shown to assist with the ability of data processing and data mining to generate more reliable and accurate outputs [23,32]. Simple methods such as standardisation of data-input requirements and prompting participants to double check improbable submissions may reduce syntactic errors in data [22]. The credibility of citizen-science data, however, will often require additional data-validation and processing strategies [63]. Professional-reference data is recommended for citizen-science programs where individual contributions must be verified [43,64], and for quantification of error in data. Professional data sets may also be integrated with citizen-science products to increase extent and completeness and to calibrate against error [9,61]. Where possible, it is also recommended that 'gold-standard' reference data sets be obtained prior to data processing, as a means of calibrating observations [43,64]. Where the citizen-science-data sample size exceeds the number of observable-data points, professional-reference data sets may not be required for data processing [34,62,65]. Inter-participant rating and reputation systems may also be used to generate reference data sets through monitoring the consistency of contributions from individuals [64].

Whether citizen-science projects directly collect in situ data and/or provide interpretation of remote-sensing images will influence strategies for deriving data of usable quality for remote-sensing calibration and validation applications. A priori determination of acceptable error rates should inform acceptable tooling for data processing and data mining, and pilot testing should seek to determine expected certainty-rate estimates [32,34]. The costs of data processing are likely to increase with the scale of data sets and associated increases in noise. Thus, a consideration is how the effectiveness of data processing tools will be evaluated and refined as citizen-science data sets grow. Applying a peer-review system to the design and products of citizen-science programs may also assist in stan-

dardising citizen-science data for use in remote-sensing calibration/validation [45]. Key program-enhancement considerations are summarised in Table 4.

**Table 4.** Data processing: Programmatic enhancement considerations.

| | |
|---|---|
| Citizen science data validation and data processing [39,43,45,64] | <ul><li>Professional-reference data is strongly recommended for citizen-science programs where individual contributions must be verified for use, and for quantification of error in data.</li><li>Interparticipant rating and reputation systems may facilitate the production of reference-data sets through monitoring the consistency of contributions from individuals</li><li>Examples of potentially useful sources of covariate information that may assist data calibration in an ongoing manner include data-capture technology used, effort information (both temporal and perceived), education history, occupation, sociodemographic characteristics, stakes and motivations in the citizen-science project, environment, training effect (actual and perceived), and confidence.</li><li>The costs of data processing are likely to increase with the scale of data sets and associate increases in noise.</li><li>Transparency with regard to data quality, certainty and potential sources of error are strongly advised for the use of citizen science.</li><li>Standardisation of strategies to communicate error and certainty estimates across data sets of different scales, or where data-mining modifications have been made, may be necessary for consistency in the use of citizen-science calibration data.</li><li>Applying a peer-review system to the design and products of citizen-science programs may assist in standardising citizen-science data to position for use in remote-sensing calibration/validation and modelling.</li></ul> |

*3.5. Guiding Questions*

Guiding questions and considerations for designing citizen-science projects to provide data supporting the remote sensing of the environment are presented in Table 5, along with associated references to support each guiding question. These guiding questions synthesise and present the current state of the literature, and provide a full life-cycle view of the key considerations associated with citizen-science projects for remote sensing. Some of these issues are highly specific to remote sensing, while others are common challenges that are broadly relevant to all citizen-science projects. However, all relevant considerations are included. For program planning and design, we recommend practitioners consider upfront the end use of data, including the need for fit-for-purpose data and data-quality requirements. We also recommend practitioners pose questions around the privacy needs of citizen-science contributors at this stage of the project. For participant engagement, practitioners should consider diversity of skills and background including training and its impact on both motivation and data quality. Questions around participant feedback and motivation should also be posed, including how this can improve data quality and the benefits of long-term participant retention for the success of the project. We recommend practitioners consider spatial and temporal scale when designing data-collection protocols, to balance the trade-off between maximising the number of samples and accounting for bias and effort. For the final stage, data processing, we recommend posing questions based on validation of the standard required for calibration and validation of citizen-science data. Finally, questions should be asked to help evaluate the data-quality elements for the use of citizen-science data.

**Table 5.** Synthesis of guiding questions for designing citizen-science projects to provide data supporting remote sensing of environment.

| Program Planning and Design: Guiding Questions and Considerations | |
| --- | --- |
| End use of data [6,30,31,34,35] | <ul><li>How will citizen-science data contribute to the calibration and validation of remote-sensing data?</li><li>What data are needed to be fit for purpose, and what type of data are possible?</li><li>What type of citizen-science data are being collected?</li><li>What are the data-quality requirements for this function?</li><li>How have stakeholders and end-users informed program design?</li></ul> |
| Transparency, trust, and privacy [20,36,37] | <ul><li>How will the need for data-quality transparency for this end use be balanced with participant privacy and the trust of both participants and data users?</li><li>Has project privacy and the data use/sharing policy been informed by survey feedback?</li></ul> |
| **Participant engagement: Guiding questions and considerations** | |
| Participant background [39,42,43] | <ul><li>Is diversity in skills and prior knowledge of participants being sought?</li><li>Is participant background likely to influence quality of contributions?</li></ul> |
| Training [20,33,38,42,45,46] | <ul><li>How will training effectiveness be evaluated?</li><li>How will training affect biases in observations? Is an over-training effect possible?</li><li>How will training affect participant motivation?</li><li>Will training effort have a significant effect on data quality?</li><li>Is the required training within the resources of the project?</li></ul> |
| Incentivisation [20,47] | <ul><li>Can participation effort be directed?</li><li>What incentivisation methods are best suited to the program?</li></ul> |
| Feedback [6,33,39,42,45] | <ul><li>Will participants be able to receive performance feedback from experts? From each other?</li><li>How will bidirectional feedback improve data quality?</li></ul> |
| Participant motivation and retention [6,33,39,42,47,49–51] | <ul><li>Does the program design and use reflect the motivation of participants?</li><li>Is there an intended optimal duration for participation?</li><li>Is long-term participant retention more important than participant numbers?</li></ul> |
| **Data collection: Guiding questions and considerations** | |
| Sampling scale and density [7,8,55–57,66] | <ul><li>Is the intended scale and density of observations operable with remote-sensing pixel resolution?</li><li>Can spatial and temporal sampling biases be mitigated through program design?</li><li>Does repeated sampling or maximising the number of samples best suit the use of data?</li></ul> |
| Structure and standardisation of sampling protocols [38,39,42,48,51] | <ul><li>Is data collection best suited to structured or unstructured sampling design?</li><li>Is standardisation of sampling protocols feasible?</li></ul> |
| **Metadata capture and technology** [38,42,59–61] | <ul><li>Is participant effort and confidence captured in data entry?</li><li>What are the minimum requirements of data-capture technology for the intended use?</li></ul> |

**Table 5.** *Cont.*

| Data processing: Guiding questions and considerations | |
| --- | --- |
| Citizen-science data validation and data processing [22,23,32–34,43,62,64,65,67] | • Is a 'gold standard' reference data set required for citizen-science calibration/validation? <br> • Can validating data be derived from citizen-science data? <br> • How will sampling design and data-capture facilitate data processing? <br> • What data-processing strategies best suit the nature of data collection? <br> • Will data-mining strategies be refined as data sets grow? <br> • How will data-quality elements be evaluated for use in citizen-science data? |

## 4. Discussion

We found that the challenges in data quality assurance faced by citizen-science programs are amplified, not in their intensity but in their importance for use in remote-sensing calibration and/or validation. While there is significant potential in synergistic co-validation of citizen science and remote sensing, optimising data interoperability between the two data sources will require consistency in data quality assurance from citizen science. In contrast to traditional in situ data collection, where protocols are systematically equipped to identify data-quality sacrifices such as spatial extent to improve reliability, the nature of remote sensing of environment presents unique challenges and requires explicitly identifying data-quality deficits and their sources. Considering the strategy undertaken by programs throughout all stages of development, implementation and analysis can be optimised to prioritise the usability of citizen-science data for remote sensing, while conserving the utility it offers (summarised in Tables 1–4).

Using our literature review as the basis, we generated a set of guiding questions for designing citizen-science projects in conjunction with remote-sensing data for environmental monitoring (Table 5). These questions can be used to help inform the design of citizen-science projects that are the most effective, efficient, and ethical, geared towards providing data to support remote sensing of the environment while maintaining citizen-science enthusiasm and participation. They were intended to help practitioners consider at what stage (or multiple stages) of a citizen-science project there is a need to help inform design to improve integration with remote sensing. While it may be unfeasible for citizen-science programs to produce complete data sets for remote-sensing applications, their usability may be improved by efforts to improve and quantify data certainty and characterise co-variates of data quality. Importantly, understanding the data quality trade-offs in citizen science upfront may assist the prioritisation of data-quality elements, while facilitating the identification of consequent reductions in the assurance of other data-quality elements.

## 5. Conclusions

This review represents the current state of the debate on citizen science for remote sensing, and serves to enhance the future development of citizen-science projects for application to remote sensing in environmental monitoring, so that the potential of citizen science can become an integral component and complementary data source [63]. Although citizen science is currently used in its widest sense in data collection rather than in scientific analysis of the data or project design, there is potential for citizens to be involved in the full sequence or workflow of a citizen-science project [6]. The role of technology (in particular) as a key enabler for the use of citizen science in remote sensing to improve data quality through data standardisation and integration with artificial intelligence will likely continue to seed its growth. In this way, citizen science can continue to develop and help meet the needs of the remote-sensing community by providing data at the spatial and temporal scales required to support the science required in the face of rapid environmental change.

**Author Contributions:** E.R., K.J.W., M.P.S., B.S. and J.L. conceived of the idea and design of the study as a research project for A.S. as part of a CSIRO internship. A.S. took the lead in the literature review, analysis and writing of the research project report in consultation with all authors. E.R. drafted the manuscript with support, input and edits from A.S., K.J.W., M.P.S., B.S. and J.L. All authors have read and agreed to the published version of the manuscript.

**Funding:** The internship for A.S. to conduct this research was funded by CSIRO's Responsible Innovation Future Science Platform.

**Acknowledgments:** We acknowledge the input received from four anonymous reviewers, who significantly improved this paper.

**Conflicts of Interest:** The authors declare no conflict of interest.

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
