# Peer review of "Use-Specific Considerations for Optimising Data Quality Trade-Offs in Citizen Science: Recommendations from a Targeted Literature Review to Improve the Usability and Utility for the Calibration and Validation of Remotely Sensed Products"

_remotesensing, doi:10.3390/rs15051407_

Round 1

Reviewer 1 Report (Previous Reviewer 2)

My concerns were well addressed. Thank you.

I think the method, in particular the queries used to search relevant papers, could be provided for greater transparency and reproducibility.

There are still some minor typos that should be addressed when preparing the camera-ready version.

Author Response

Reviewer 2 Report (Previous Reviewer 3)

This paper analyze how the growth of citizen science can be used as a valuable source data for remote sensing. The authors analize the mismatch between in situ data quality requirements for remote sensing product development and current data quality assurance in citizen science. For this, the authors conducted a targeted literature review into the nature of data quality issues faced by citizen science projects for routine incorporation into terrestrial environmental monitoring systems. 

From my point of view, the topic is interesting, and I support the authors to continue researching the topic. However, after reading the article, it gives me the idea that his contribution is not significant. The guiding questions and considerations obtained and presented in table 1 for designing citizen science projects to provide data supporting remote sensing of environment was obtained from several authors, and it is interesting and valuable information.

However, the problem I find is that the authors don't really make a significant contribution in the area. There is no statistical analysis, no model, no clear methodology to follow. After reading the article, although it is quite informative, it has no scientific contribution from my point of view. For example, the Guiding questions and considerations should have been tested with a group of experts in the area to analyze if they are well structured, and if they are really valuable or not, together with statistical results that support this information. The results of that study would be somewhat more valuable from my point of view if some kind of data analysis were presented. Additionally, the paper is not really a survey, which indicates all the work carried out to date, and which puts them in tables, taking out the most important areas of work, representing it in graphs, etc. The paper rather proposes a guide for data collection, without having any tangible results beforehand.

From my point of view, this paper is not ready to be published in this journal due to its lack of content and scientific and technical contribution.

Author Response

Reviewer 3 Report (New Reviewer)

The manuscript is devoted to the important issue of involving citizen scientists in data collection for remote sensing cal/val and utilizing of collected data. Authors reviewed impressive mount of literature and collected recommendations from the sources.

The manuscript is supposed to help a person/team who consider to organize a citizen science campaign and/or utilize citizen science data for cal/val purposes. It highlights the limitation of this source of data and issues which requires special attention. I found the paper useful to get a first impression on the topic, however I miss concrete examples and recommendations. The statements are typically too vague and not really useful, e.g. “Participant training presents a useful tool …, however, overtraining can reduce quantity”, “Gamification … may present useful tools” but “may also risk disincentivising” (Section 3.2).

The “calibration/validation” are typically mentioned together (e.g. Page 3. Section 3.1), while the requirements these two tasks are substantially different. Citizen science data might meet requirements of calibration data, but be not suitable for the validation.

I also suggest authors to have a look at the recent paper by See et al. (2022) Lessons learned in developing reference data sets with the contribution of citizens … https://doi.org/10.1088/1748-9326/ac6ad7

Author Response

Reviewer 4 Report (New Reviewer)

The authors give a targeted literature review of how crowdsourcing projects processing remotely sensed images in environmental sciences can assure data quality. The authors stated that the literature review is targeted, although such paper is missing such as Comber, A., See, L., Fritz, S., Van der Velde, M., Perger, C., & Foody, G. (2013). Using control data to determine the reliability of volunteered geographic information about land cover. International Journal of Applied Earth Observation and Geoinformation23, 37-48.

Based on the literature review the authors developed a set of guiding questions for crowdsourcing project design.

It would also be beneficial if the authors would give the crowdsourcing project designer a table (similar to table 1) containing the tools for enhancing the quality of such a project and the possible pitfalls they should be careful of.

Author Response

This manuscript is a resubmission of an earlier submission. The following is a list of the peer review reports and author responses from that submission.

Round 1

Reviewer 1 Report

The paper is interesting and well written.

It provides a useful and important insight into issues related to citizen science.

I suggest the term "paid" professionals be replaced with discipline experts as not all scientist are paid to undertake data collection and validation.

Much of the remote sensing data could be considered as spatial or geographic information.  A suggestion is that the term volunteered geographic information (VGI) would also provide another suitable source of literature to build on to address and support the guiding questions.

The role of technology is touched on through the data collection but this is perhaps a key enabler for to improve data quality through, data standardisation, supporting geo-tagged images and use and support of of AI.

Reviewer 2 Report

This paper provides a review on the nature of data quality issues related to citizen science projects with the idea of incorporating the collected data into terrestrial environmental monitoring systems. The topic is interesting and pertinent and the paper is well-written. However, some clarifications are needed and missing components need to be addressed.

In my opinion, the authors should briefly discuss the meaning of Citizen Science, which is different from other concepts such as crowdsourcing or Volunteered Geographic Information. The term citizen science is often used with slightly different meanings and it should be clear which definition is being used in the scope of this review.

The methodology needs to be improved as it contains some negative aspects:

i) the Google scholar database usually brings also non-scientific and non-indexed documents and doesn't seem to be the best source for a journal review;

ii) as it is described it is impossible to reproduce and get similar results as the authors;

iii)  The search query/queries should be provided, again for reproducibility purposes. Having only the key search terms is not enough;

iv) The method for removing non-relevant papers is also not clear. More details should be provided to make the methods stronger and more transparent;

Regarding the results, a few questions should be addressed and clarified in the paper:

i) how did the authors end up with the four critical elements used in the document? The paper does describe the respective method/analysis.

ii) In my view, the authors should clearly demonstrate how they extracted the questions from the analysis of papers. The questions should be presented in the text in the first place, and the table should be presented at the end of the results section as a kind of summary of relevant content. This would make the review more interesting and clear. It is also not clear how readers should use these questions to improve citizen science projects.

This sentence [lines 55-58] needs to be rewritten to clarify its meaning. “Funding limitations result in professionals needing to compromise on re-visit frequency (Sparrow et al, 2020b), hence longer revisit times, and the need to integrate these techniques with remote sensing to quantify rapid change and inform management decisions (Sparrow et al, 2020a).”

Reviewer 3 Report

The Authors performed a literature review related to the nature of data quality issues faced by citizen science projects for the Incorporation of terrestrial environmental monitoring systems. From this review, the authors identified four elements of citizen science program design for optimal data use in remote sensing calibration and validation: (1) Program Planning and Design; (2) Participant engagement; (3) Data Collection; and (4) Data Processing. The authors also present a Trade-Off to inform the best practice implementation of data quality assurance in citizen science projects.

In my sight, the paper is well structured and written. Some minor details in the writing have to be corrected. However, the article's contribution is not clear from my point of view. The section of materials and methods section is presented in only a few lines. The scientific contribution of this paper is not enough for a journal paper from my criteria. In addition, I leave the following comments.

- Change the title. It is assumed that the Paper is a revision of literature, and this is not noticeable in the title.

- The contribution of the article should be focused on reviewing exhaustive literature. The number of articles, the keywords that were used in the search, the exclusion criteria for the final selection of relevant articles, and the selected items must be carefully performed and clearly indicated in a table.

- The section of materials and methods section is presented in only a few lines. What is the scientific contribution of the article?

- The contributions of the article in the introduction section must be clearly indicated.

- The recommendations to calibrate and validate the sensors that the authors make are not based on experiments that they have made. This could make the author's suggestions subjective. Discuss this in detail in the discussion section.

- I recommend that the article have sections of discussion and conclusions separately.